# Need Satisfaction and Need Thwarting in Physical Education and Intention to be Physically Active

**Ricardo Cuevas-Campos [1],\*, Juan Gregorio Fernández-Bustos [2] , David González-Cutre [3] and Andrea Hernández-Martínez [1]**

1   Department of Didactics of Musical, Plastic and Corporal Expression, Faculty of Education of Ciudad Real, University of Castilla-La Mancha, 13071 Ciudad Real, Spain; Andrea.hernandez@uclm.es
2   Department of Didactics of Musical, Plastic and Corporal Expression, Faculty of Education of Albacete, University of Castilla-La Mancha, 02071 Albacete, Spain; Juang.fernandez@uclm.es
3   Department of Sport Sciences, Sport Research Center, Miguel Hernández University of Elche, 02302 Elche, Spain; dgonzalez-cutre@umh.es
\*   Correspondence: Ricardo.Cuevas@uclm.es

**Abstract:** The aim of this study is to evaluate a theoretical model for analyzing the influence of need satisfaction, need thwarting, motivation, enjoyment, boredom, and exhaustion in physical education on the intention to be physically active. In addition, we examined the mediation effect of motivation between basic psychological needs and the intention to be physically active. The study was based on self-determination theory. To achieve this, 480 students between 12 and 18 years old filled out a questionnaire to measure the satisfaction and thwarting of basic psychological needs, motivation, enjoyment, boredom, exhaustion, and intention to be physically active. The results of path analysis showed the relevance of the satisfaction of the need for competence in physical education in order to increase self-determined motivation, enjoyment, and intention to be physically active. Additionally, intention was positively predicted by enjoyment and negatively predicted by exhaustion. Need thwarting directly predicted negative consequences, such as boredom and exhaustion. Motivation mediated the relationship between basic needs and intention to be physically active. These data highlight the importance of considering basic psychological need thwarting in studies on the promotion of physical activity.

**Keywords:** adolescence; self-determination; physical activity; path analysis; motivation

## 1. Introduction

According to the World Health Organization [1], the regular practice of physical activity is associated with extensive health benefits, especially during adolescence [2]. Ramos, Rivera, Moreno, and Jimenez-Iglesias [3] found an increase in sedentary behaviour and lower levels of perceived health from a lack of physical activity in adolescents. For this reason, physical education (PE), understood in this case as physical education provided in compulsory education, can be fundamental for the promotion of physical activity and healthy habits among young people [4–6]. Accordingly, the present study examines how some psychological variables could influence the intention to practice physical activity in adolescents, that is, how psychological variables can contribute to adherence to the practice of physical activity in adolescence. The aim of this was to suggest PE interventions that could effectively promote the long-term practice of physical activity.

According to the theory of planned behaviour [7], the intention to perform a particular behaviour is the variable that best predicts its effective execution [8]. Several studies on adolescents have demonstrated the predictive capacity of the intention to be physically active, with regards to



the implementation of such behaviour [9,10]. In addition, recent studies (e.g., [11,12]) have shown the usefulness of self-determination theory (SDT) [13,14] in explaining the reasons for physical activity to be planned in the future. SDT indicates the existence of different types of motivation. From the highest to the lowest degree of self-determination, we can find various levels of regulation, such as: intrinsic motivation (based on interest and enjoyment in participating in the activity), integrated regulation (when the activity is integrated into a lifestyle), identified regulation (when the conduct is considered useful to the individual), introjected regulation (grounded on the consequences of and feelings of guilt from failing to perform the behaviour), external regulation (when the behaviour is performed for external recompenses), and amotivation (absence of motivation). Several PE works have linked intrinsic and identified regulations with positive variables, such as interest, self-esteem, effort and vitality. Oppositely, the external regulation and introjected regulation, along with amotivation, have been related with negative consequences, such as boredom and unhappiness [15]. Moreover, more self-determined forms of motivation predict both the intention to be physically active, and the actual behaviour [16,17].

SDT indicates that individuals have basic psychological needs, which must be fulfilled to ensure good psychological functioning in people. There are three needs: autonomy, competence and relatedness. According to Deci and Ryan [18], competence represents the level of mastery that a person feels when performing a task. The need for autonomy refers to a person's ability to choose, and their involvement in the process. Finally, the need of relatedness is based on the perception of a connection with others, and acceptance by other people. The satisfaction of these needs is related to benefits such as greater self-determined motivation, vitality, and positive affect [19–23], as well are increased physical exercise [24,25]. In the context of PE [26], the satisfaction of the need for competence has been demonstrated to be the most important factor [27], and is directly related to the intention to practice physical activity in adolescents [28–30]. However, a negative environment can also thwart these needs. Bartholomew, Ntoumanis, Ryan and Thørgersen-Ntoumani [31] developed the concept of psychological need thwarting, referring to the negative state that is experienced when an individual perceives that his or her psychological needs are being actively denied by the actions of others. Several studies have associated psychological need thwarting with negative variables such as burnout [32], negative affect [21], physical/social anxiety [33], boredom [34] and controlled motivation and amotivation [35] in youth.

There are other psychological variables related to the types of motivation and psychological needs that can exert a direct influence on the intention to be physically active. Enjoyment in PE has been associated with more self-determined motivation types, and with the satisfaction of competence [36,37]. In contrast, boredom in class has been positively predicted by variables such as external regulation and amotivation [38], and negatively predicted by intrinsic motivation [39]. On the other hand, physical and emotional exhaustion, such as that which occurs in burnout, can also influence the intention to be physically active. Thus, burnout has been negatively associated with self-determined forms of motivation, and with the intention to practice in young athletes [40].

Despite the advances described, there is a need to explore new predictor variables to more effectively encourage the practice of physical activity [17]. Previous studies have been based on predictive models with a limited number of variables (e.g., [41]). Hence, the aim of this study was to test a theoretical model based on the predictive capacity of satisfaction and thwarting of basic psychological needs with regards to motivational regulations. In turn, the predictive capacity of motivation with regards to enjoyment, boredom, and exhaustion was analysed. As a result, we looked into the predictions of these three variables with regards to the intention to be physically active. Following the hierarchical model of motivation [42] as a reference, the following hypotheses were considered: (1) The satisfaction of basic psychological needs will positively predict more autonomous forms of motivation (intrinsic and identified) and negatively predict the remaining regulation styles and amotivation. Psychological need thwarting will negatively predict intrinsic and identified motivation, and positively predict the rest. (2) In turn, intrinsic and identified motivation will positively predict enjoyment and negatively predict exhaustion and boredom; introjected and external regulation

and amotivation will negatively predict enjoyment and positively predict exhaustion and boredom. (3) Enjoyment will predict intention to be physically active, whereas it will be negatively predicted by exhaustion and boredom. Finally, (4) the different types of motivation will mediate the relationships between basic needs and the intention to be physically active.

## 2. Materials and Methods

### 2.1. Participants

The participants were 480 Spanish Caucasian students (265 girls and 215 boys) between 12 and 18 years of age (M age = 14.63, SD = 1.79), who were studying PE in secondary compulsory education selected by convenience of two public schools of the region of Castilla-La Mancha (Albacete, Spain).

### 2.2. Procedure

For the data collection, the objective of the study was described to the students, the teachers, the school principals, and the parents of the participants. This study was approved by an institutional committee for research and ethics. Additionally, the necessary permissions were obtained from the participants and from a Spanish university. The study was voluntary and anonymous. The questionnaires were completed in PE classes, taking approximately 25 min. A researcher explained the instructions and answered any questions raised by the participants.

### 2.3. Measures

Psychological need satisfaction. The Basic Psychological Needs in Exercise Scale (BPNES; [43]), in its Spanish form [44] was used. The stem "In my PE classes . . . " was follow by four items to assess each of the psychological needs: autonomy satisfaction (e.g., "I feel that the way I exercise is definitely an expression of myself"), competence (e.g., "I feel that exercise is an activity in which I do very well") and relatedness (e.g., "I feel very much at ease with the other exercise participants"). Each of the three dimensions consisted of four items, and responses were specified on a 5-point scale varying from 1 (strongly disagree) to 5 (strongly agree). Moreno et al. [43] found adequate reliability (Cronbach's alphas coefficients above 0.78) and validity ($\chi^2$ = 36.19, comparative fit index (CFI) = 0.94, root mean square error of approximation (RMSEA) = 0.07) for this scale.

Psychological need thwarting. The Spanish version [45] of the Psychological Need Thwarting Scale (PNTS; [31]) was adapted to the PE context. It was constituted by the stem "In my PE classes . . . " and by 12 items (four for each need). The items assessed autonomy thwarting (e.g., "I feel pushed to behave in certain ways"), competence thwarting (e.g., "There are times when I am told things that make me feel incompetent") and relatedness thwarting (e.g., "I feel other people dislike me"). Responses were collected on a 5-point scale beginning from 1 (strongly disagree) to 5 (strongly agree). Cuevas et al. [45] found suitable reliability (Cronbach's alphas coefficients above 0.80) and validity ($\chi^2$ = 248.61, CFI = 0.95, RMSEA = 0.08) for this scale.

Motivation. The Scale to Assess the Motivation in PE Classes [46] was used. The stem "Why do you take part in PE classes?" was follow by 20 items (four for each subscale) to assess intrinsic motivation (e.g., "because Physical Education is fun"), identified regulation (e.g., "Because I can learn skills that I could use in other areas of my life"), introjected regulation (e.g., "Because I feel bad if I don't participate in the activities"), external regulation (e.g., "To show the teacher and peers my interest in the subject") and amotivation (e.g,. "I don't know clearly; because I don't like the subject at all"). Following the SDT structure, intrinsic motivation and identified regulation were collapsed under a variable named "autonomous motivation". Introjected and external regulations were merged into a dimension named "controlled motivation". The responses were provided on a 5-point scale ranging from 1 (strongly disagree) to 5 (strongly agree). Sánchez-Oliva et al. [46] found adequate reliability

(Cronbach's alphas coefficients above 0.77) and validity ($\chi^2$ = 473.64, CFI = 0.96, RMSEA = 0.05) for this scale.

Enjoyment and boredom. The Spanish version adapted for PE [47] of the Sport Satisfaction Instrument [48] was used. Two factors were evaluated: satisfaction–enjoyment with five items (e.g., "I typically have fun in PE classes"), and boredom with three items (e.g., "In PE classes, I typically get bored"). The responses were provided on a 5-point scale ranging from 1 (strongly disagree) to 5 (strongly agree). Baena-Extremera et al. [47] found adequate reliability (Cronbach's alphas coefficients above 0.79) and validity ($\chi^2$ = 38.53, CFI = 0.99, RMSEA = 0.03) for this scale.

Exhaustion. To assess exhaustion in PE classes, a factor of the Athlete Burnout Questionnaire (ABQ; [49]) was adapted from the Spanish version [50]. The items referring to the sport modality were customized for PE classes. This factor was composed of five items (e.g., "I am exhausted due to the physical and mental demands of the PE class"). The responses were provided on a 5-point scale ranging from 1 (strongly disagree) to 5 (strongly agree). De Francisco et al. [50] found adequate reliability (Cronbach's alphas coefficients above 0.79) and validity ($\chi^2$ = 179.49, CFI = 0.91, RMSEA = 0.08) for this scale.

Intention to be physically active. The Spanish version [51] of the Intention to be Physically Active Scale [52] was used. The scale was composed of five items (e.g., "Outside PE lessons I like to do sport"). The responses were provided on a 5-point scale ranging from 1 (strongly disagree) to 5 (strongly agree). Moreno et al. [51] found adequate reliability (Cronbach's alpha coefficient of 0.94) and validity ($\chi^2$ = 72.77, CFI = 0.98, RMSEA = 0.05) for this scale.

## 2.4. Data Analysis

The means, standard deviations, and bivariate correlations were calculated. Validity was confirmed with a confirmatory factorial analysis (CFA) of the measurement model. For testing reliability, average variance extracted (AVE), Cronbach's alpha, and composite reliability (CR) were calculated. AVE specifies the variance of the items taken by the latent construct in contrast with the variance captured by measurement error; results higher or equal to 0.5 for AVE are considered adequate. A value above 0.7 for Cronbach's alpha shows adequate internal consistency of the scale items [53]. CR indicates the consistency of the observed variables with the measurement latent construct; values higher or equal to 0.7 for CR are considered acceptable. The IBM SPSS Amos 24.0 programs were used.

Due to the large number of variables, we opted to use path analysis to test the model. The distribution of the sample data was non-normal. In cases with a lack of normality in the data, the procedure of bootstrapping affords robust standard errors estimated [54]. Hence, the maximum-likelihood estimation method with bootstrapping was used for the absence of normality of the sample data. For interpretation of the model fit, some indexes were used: the chi-square value, the comparative fit index (CFI), the Tucker-Lewis index (TLI), and the root mean square error of approximation (RMSEA). The fit of the model could be considered satisfactory if CFI and TLI are at least of 0.95, and if RMSEA is lower than or equivalent to 0.08 [55–57]. As well as this, a confidence interval (e.g., 90%) was produced to indicate the level of the RMSEA precision; an adequate fit occurs if the upper limit is lower than or equal to 0.08, and if the complete range is smaller than 0.05 [58]. Mediation analysis was implemented by the bootstrapping technique [59]. This technique generates a confidence interval for indirect effects: if zero is included in the confidence interval, the indirect effect of the mediator variable is considered non-significant.

## 3. Results

### 3.1. Preliminary Analysis

The fit indices obtained in the confirmatory factor analysis for the different measuring instruments (Table 1) were acceptable. Table 2 presents the correlations, means, standard deviations, Cronbach's alphas, composite reliability, and average variance extracted values for each dimension. The values were

suitable for all constructs, except boredom, whose value can be considered minimally acceptable [60]. Based on the theoretical foundations, the correlations between the different variables were standard.

**Table 1.** Confirmatory Factor Analysis.

|  | $\chi^2/df$ | CFI | TLI | RMSEA | $p$ |
|---|---|---|---|---|---|
| Need Satisfaction | 3.46 | 0.95 | 0.94 | 0.07 | 0.00 |
| Need Thwarting | 4.88 | 0.93 | 0.91 | 0.08 | 0.00 |
| Motivation | 3.07 | 0.94 | 0.93 | 0.06 | 0.00 |
| Enjoyment and Boredom | 1.90 | 0.99 | 0.98 | 0.04 | 0.01 |
| Exhaustion | 1.18 | 0.99 | 0.99 | 0.02 | 0.00 |
| Intention to be physically active | 4.79 | 0.98 | 0.96 | 0.08 | 0.01 |

$\chi^2/df$ = chi-squared by degrees of freedom ratio; CFI = comparative fit index; TLI = Tucker-Lewis index; RMSEA = root mean square error of approximation.

### 3.2. Path Analysis

First, we tested the hypothesized model. As noted above, the model reflected that need satisfaction positively predicted autonomous regulations and negatively predicted controlled regulations and amotivation. Need thwarting negatively predicted self-determined motivational regulations and positively predicted controlled regulations and amotivation. Additionally, the autonomous forms of motivation positively predicted enjoyment and negatively predicted exhaustion and boredom, while exhaustion and boredom was positively predicted by introjected and external regulations and amotivation, and enjoyment was negatively predicted by introjected and external regulations and amotivation. In turn, exhaustion and boredom negatively predicted the intention to be physically active, while enjoyment positively predicted intention to be physically active. Several of the proposed relationships were not significant, and the fit indices for this initial model were not adequate: $\chi^2$ (df) = 791.15 (42), $p < 0.001$, CFI = 0.83, TLI = 0.58, RMSEA (IC 90) = 0.193 (0.181–0.205). Thus, the model was revised. To do so, the modification indices were examined. These indices showed the need to include in the model a direct relationship between competence satisfaction and enjoyment, exhaustion, and intention. Satisfaction of relatedness was linked with enjoyment and boredom. A direct relationship between competence need thwarting and exhaustion was included. From these findings, it was assumed that the following relationships could be identified: autonomy need thwarting was directly linked with boredom and exhaustion, and relatedness need thwarting were directly associated with boredom. All of these new relationships were theoretically plausible. Thus, the resulting final model (Figure 1) yielded adequate fit indices: $\chi^2$ (df) = 46.966 (21), $p < 0.001$, CFI = 0.99, TLI = 0.98, RMSEA (CI 90) = 0.051 (0.031–0.070).

**Table 2.** Descriptive Statistics, Reliability Estimates and Pearson Correlations.

| Variable | *M* | *SD* | α | CR | AVE | 1 | 2 | 3 | 4 | 5 | 6 | 7 | 8 | 9 | 10 | 11 | 12 | 13 | 14 |
|---|---|---|---|---|---|---|---|---|---|---|---|---|---|---|---|---|---|---|---|
| 1. Competence Sat. | 3.64 | 0.78 | 0.77 | 0.76 | 0.50 | | | | | | | | | | | | | | |
| 2. Autonomy Sat. | 3.43 | 0.86 | 0.75 | 0.77 | 0.51 | 0.78 ** | | | | | | | | | | | | | |
| 3. Relatedness Sat. | 4.09 | 0.85 | 0.87 | 0.87 | 0.62 | 0.60 ** | 0.50 ** | | | | | | | | | | | | |
| 4. Competence Fru. | 2.01 | 0.89 | 0.79 | 0.79 | 0.51 | −0.27 ** | −0.18 ** | −0.36 ** | | | | | | | | | | | |
| 5. Autonomy Fru. | 2.19 | 0.90 | 0.77 | 0.78 | 0.50 | −0.18 ** | −0.17 ** | −0.23 ** | 0.72 ** | | | | | | | | | | |
| 6. Relatedness Fru. | 1.76 | 0.99 | 0.79 | 0.82 | 0.55 | −0.21 ** | −0.12 ** | −0.42 ** | 0.76 ** | 0.57 ** | | | | | | | | | |
| 7. Intrinsic | 3.90 | 0.86 | 0.85 | 0.85 | 0.57 | 0.67 ** | 0.65 ** | 0.52 ** | −0.17 ** | −0.16 ** | −0.15 ** | | | | | | | | |
| 8. Identified | 3.74 | 0.88 | 0.83 | 0.84 | 0.55 | 0.66 ** | 0.60 ** | 0.44 ** | −0.22 ** | −0.21 ** | −0.19 ** | 0.68 ** | | | | | | | |
| 9. Introjected | 2.70 | 0.86 | 0.78 | 0.81 | 0.56 | 0.25 ** | 0.34 ** | 0.18 ** | 0.07 ** | 0.12 ** | 0.02 | 0.31 ** | 0.28 ** | | | | | | |
| 10. External | 3.05 | 1.04 | 0.74 | 0.74 | 0.49 | 0.34 ** | 0.34* | −0.21 ** | 0.07 ** | 0.09 * | 0.07 | 0.39 ** | 0.34 ** | 0.56 ** | | | | | |
| 11. Amotivation | 1.64 | 0.87 | 0.80 | 0.84 | 0.52 | −0.41 ** | −0.37 ** | −0.41 ** | 0.41 ** | 0.42 ** | 0.38 ** | −0.52 ** | −0.47 ** | 0.01 | −0.05 ** | | | | |
| 12. Enjoyment | 4.00 | 0.89 | 0.89 | 0.89 | 0.62 | 0.72 ** | 0.66 ** | 0.63 ** | −0.24 ** | −0.22 ** | −0.23 ** | 0.81 ** | 0.65 ** | 0.26 | 0.34 ** | −0.55 ** | | | |
| 13. Boredom | 1.87 | 0.96 | 0.65 | 0.65 | 0.46 | −0.46 ** | −0.42 ** | −0.47 ** | 0.40 ** | 0.40 ** | 0.39 ** | −0.54 ** | −0.34 ** | −0.10 * | −0.14 ** | 0.66 ** | −0.65 ** | | |
| 14. Exhaustion | 1.90 | 0.86 | 0.89 | 0.89 | 0.62 | −0.33 ** | −0.21 ** | −0.29 ** | 0.54 ** | 0.50 ** | 0.43 ** | −0.24 ** | −0.22 ** | 0.09* | 0.05 | 0.40 ** | −0.34 ** | 0.39 ** | |
| 15. Intention | 3.90 | 0.97 | 0.86 | 0.86 | 0.55 | 0.63 ** | 0.44 ** | 0.43 ** | −0.19 ** | −0.10 ** | −0.13 ** | 0.54 ** | 0.49 ** | 0.19 ** | 0.20 ** | −0.37 ** | 0.61 ** | −0.40 ** | −0.34 ** |

* $p < 0.05$, ** $p < 0.01$. Note. α = Cronbach's alpha; CR = Reliability composite; AVE = Average variance extracted.

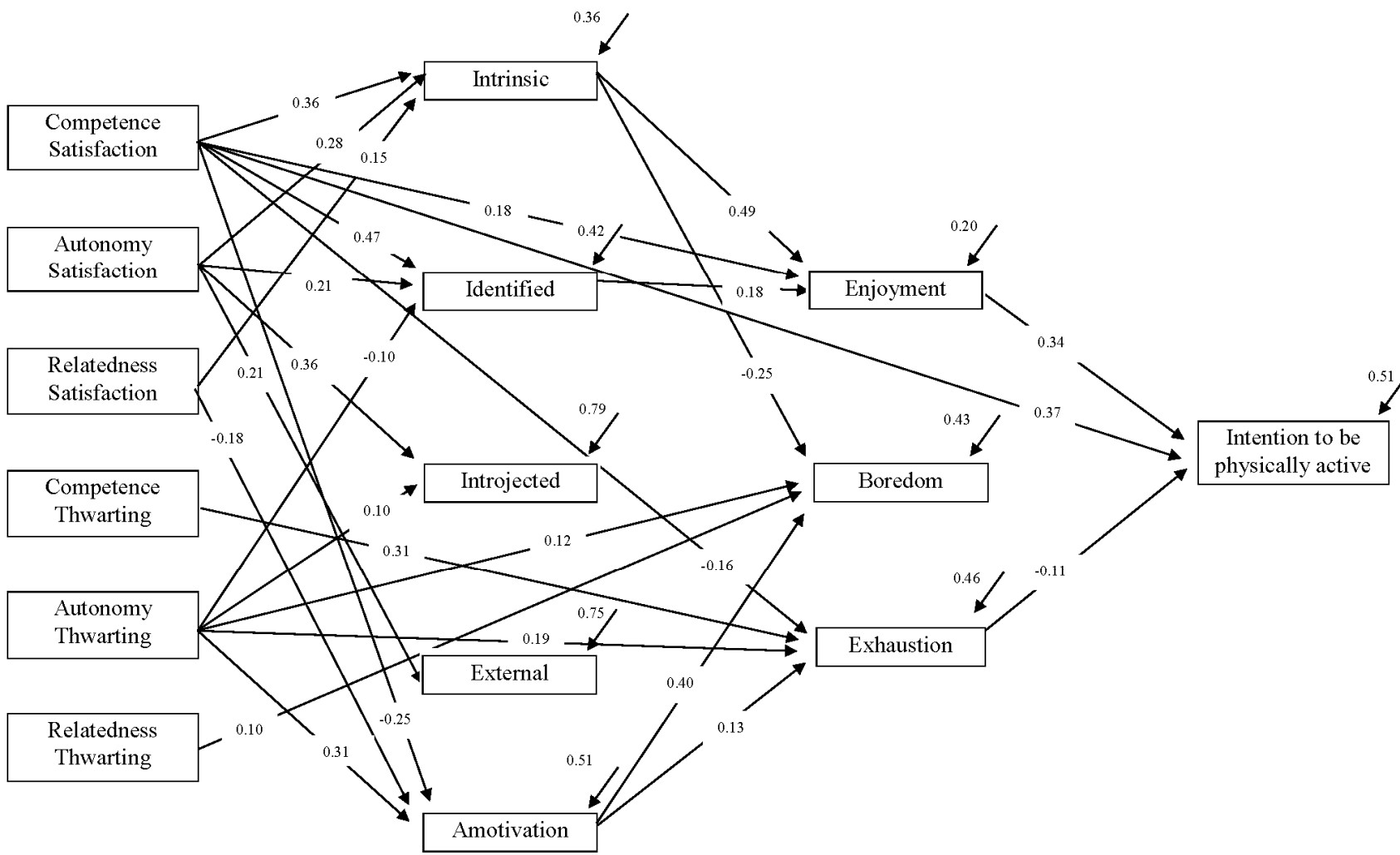

**Figure 1.** Final model for the prediction of intention to be physically active in secondary students. Note: Standardized path coefficients and residuals are presented. Non-significant parameters and correlations coefficients between exogenous variables are not presented for presentation simplicity purposes.

### 3.3. Mediation Analysis

Table 3 shows the indirect effects of basic need satisfaction and thwarting on the intention to be physically active through the different types of motivation (autonomous, controlled, and amotivation). The total indirect effects were significant ($p < 0.05$). The specific indirect effects for need satisfaction indicate that these effects were not significant for all cases of controlled motivation and amotivation. In contrast, autonomous motivation mediated the positive indirect effect of basic need satisfaction on the intention to be physically active ($p < 0.05$). When analyzing specific indirect effects of need thwarting, it was detected that these effects were not significant for all cases of autonomous and controlled motivation. Oppositely, the negative indirect effect of need thwarting on intention was mediated by amotivation ($p < 0.05$).

**Table 3.** Standardized Indirect Effects. Dependent variable: Intention.

| | | Specific Indirect Effects | | |
|---|---|---|---|---|
| Independent Variable | Total Indirect Effect (95% CI) | Autonomous Motivation (95% CI) | Controlled Motivation (95% CI) | Amotivation (95% CI) |
| Competence Satisfaction | 0.25 * (0.20 to 0.31) | 0.23 * (0.18 to 0.28) | 0.01 (−0.01 to 0.03) | 0.01 (−0.02 to 0.03) |
| Autonomy Satisfaction | 0.32 * (0.27 to 0.39) | 0.29 * (0.23 to 0.35) | 0.03 * (0.01 to 0.05) | 0.01 (−0.01 to 0.04) |
| Relatedness Satisfaction | 0.21 * (0.16 to 0.29) | 0.14 * (0.08 to 0.23) | 0.01 (−0.01 to 0.03) | 0.05 * (0.02 to 0.09) |
| Competence Thwarting | −0.11 * (−0.18 to −0.05) | −0.01 (−0.07 to 0.07) | 0.02 * (0.01 to 0.05) | −0.13 * (−0.18 to −0.09) |
| Autonomy Thwarting | −0.15 * (−0.22 to −0.08) | −0.04 (−0.11 to 0.03) | 0.03 * (0.01 to 0.05) | −0.14 * (−0.19 to −0.10) |
| Relatedness Thwarting | −0.13 * (−0.22 to −0.07) | −0.01 (−0.09 to 0.07) | 0.01 (0.01 to 0.04) | −0.14 * (−0.20 to −0.09) |

Note. CI = 95% Confidence Intervals, * = CI does not include zero.

## 4. Discussion

The main objective of this study was to test a model of variables associated with the intention of being physically active in PE students in secondary education, using SDT [13] as a framework. Specifically, in the model, we explored the predictive capacity of satisfaction and thwarting of basic psychological needs with regards to the self-determination types of motivation; in turn, we tested the predictions of motivation types regarding enjoyment, boredom, and exhaustion. Finally, we studied the predictive capacity of enjoyment, boredom, and exhaustion with regards to the intention to be physically active. In general, the main objective was to obtain a final model to clarify whether the relationships of the proposed variables have a notable impact on the intention to practice physical activity. In addition, we studied the mediation effects of motivation between basic psychological needs and the intention to be physically active.

The first hypothesis, referring to the predictive capacity of basic psychological needs, was partially confirmed. In line with previous work, the satisfaction of competence positively predicts self-determined forms of motivation [22,23,29], and negatively predicts amotivation [23]. In addition, the satisfaction of competence exerts a positive influence on enjoyment and intention to be physically active. Contrarily to this, in line with Li et al. [61], satisfaction of competence has a negative relationship with exhaustion. These data are in line with the findings of several studies that highlight the importance of promoting the satisfaction of competence in people to promote active lifestyles, given that this perception encourages enjoyment [36,62] and favours the practice of physical activity [16,17,28,29,51]. According to Ntoumanis [27], the satisfaction of competence seems particularly important in PE, because of its direct relationship with both motivational regulations and with other important variables regarding the intention to be physically active. Autonomy satisfaction positively predicts self-determined motivation, according to the findings of previous studies [23,37]; however, contradicting the hypothesis, it also positively predicts introjected and external regulation. This relationship was also found by Standage et al. [23], which can be explained by considering PE classes to be mandatory, and very heterogeneous in terms of students and their types of motivation [63]. Consequently, many students will participate only to avoid feelings of guilt (introjected regulation) or punishment by teachers (external regulation). Therefore, it is not surprising that, while high satisfaction with autonomy in classes is found, they also obtain high scores for non-self-determined forms of regulation. The satisfaction in

relatedness positively predicts intrinsic motivation, in accordance with the findings of Moreno et al. [51]. In addition, it negatively predicts amotivation, underlining the importance of maintaining social links in class. However, satisfaction in relatedness seems to be the least relevant of the three psychological needs in PE [12], considering that its regression weights are low, and it only predicts the motivational regulations that are located at the poles of the self-determination continuum.

Psychological need thwarting has low predictive capacity with regards to the types of motivation, in that only the thwarting of autonomy positively predicts amotivation and introjected regulation, and negatively predicts identified regulation. Autonomy thwarting also predicts boredom and exhaustion of students. Therefore, to promote student motivation and engagement, the importance of offering students some autonomy to make decisions in the management of the class is highlighted. Given that PE is a compulsory activity, it is particularly important that teachers do not thwart students' autonomy any more than the compulsory context does. This could explain why the thwarting of autonomy was the need that best predicted maladaptive variables in the present study. Additionally, the positive prediction of thwarting of competence with regards to exhaustion, and the positive prediction of thwarting of relatedness with regards to boredom, are highlighted. Although there is not much evidence related to these variables, these results are consistent with the findings of previous studies with athletes [34] and PE students [35]. Additionally, these data are consistent with other studies that have associated need thwarting with non-self-determined motivational regulation and burnout in athletes [31,32] and teachers [45,64,65]. An important conclusion of this study is that, in general, need thwarting is more directly related to certain consequences (boredom, exhaustion) than motivational regulations.

The second hypothesis was also partially confirmed. Many hypothesized associations were not confirmed in the final model. Enjoyment was positively predicted by intrinsic motivation. Furthermore, intrinsic motivation negatively predicted boredom. These results are consistent with the findings of previous studies [36,39]. Moreover Kalajas-Tilga, Koka, Hein, Tilga and Raudsepp [66] found a positive relationship between intrinsic motivation and psychological needs in 6th–8th grade PE students. Furthermore, in line with self-determination foundations, enjoyment is also positively predicted by identified regulation, and exhaustion and boredom is predicted by amotivation.

Similarly, the third hypothesis was partially supported. Enjoyment was confirmed as a positive predictor of the intention to practice, in line with previous studies on PE [67,68]. On the other hand, exhaustion negatively predicted intention to be physically active, which is consistent with previous research that has found an inverse association between exhaustion and the practice of physical exercise [69].

The fourth hypothesis predicted that the different dimensions of motivation mediated the relation between basic needs and intention. This assumption was just in part confirmed. The total indirect effects of basic needs on intention were significant. However, the exploration of the specific indirect effects for need satisfaction showed that just autonomous motivation played a mediating role in the relationship between needs and intention. On the other hand, the analysis of the specific indirect effects of need thwarting showed that only amotivation played a mediating role in the relationship between needs and intention. These effects suggest that basic needs satisfaction results in higher levels of intention via increased autonomous motivation. Additionally, basic needs thwarting results in lower levels of intention via increased amotivation. Such results are especially helpful because they provide evidence of the psychological mechanisms through which basic needs can influence adolescents' intentions to be physically active.

## 4.1. Applied Implications

The results show that the thwarting of students' needs by external agents could cause a sense of boredom, exhaustion, and amotivation. This model helps to rank the variables that predict intention to be physically active. The variable with greatest direct influence on the intention to be active is the satisfaction of competence. In addition, satisfaction of competence has a significant influence on

variables that are associated with the positive development of adolescents, such as self-determined motivation and enjoyment. Thus, in order to promote physical activity through PE, it is necessary to prioritize interventions that are autonomy-supportive and less controlling [70–72], to improve students' sense of efficacy, stressing its importance above other variables such as autonomy or enjoyment. Additionally, the model suggests that teachers should approach the teaching of PE by trying to avoid the thwarting of the needs of their students in order to promote physical activity. To avoid the thwarting of competence, we suggest avoiding the use of negative feedback focused exclusively on errors, and the assignment of tasks that are not adapted to the skill of each student. To avoid the thwarting of autonomy, the teacher should not ignore students' views and suggestions. Finally, to avoid the thwarting of relatedness, it is important that the students do not feel that the teacher does not care about them.

*4.2. Limitations and Future Directions*

This study has some limitations that should be considered in the design of future studies. The predictive capacities of some variables with regards to the intention to practice physical activity have been studied, but not their predictive capacity with regards to real behaviour. Although intention has predicted behaviour in some studies [11,12], it would be useful to conduct studies that support the influence of the variables analysed on the effective performance of physical activities in students. Furthermore, in regards to extrapolating these results, the selection of the sample (one country, one educational level) suggests caution is required. Consequently, it is recommended that the generalizability of this model be tested in other countries and academic levels.

## 5. Conclusions

This article provides evidence for the theoretical framework of the promotion of physical activity, focusing on the satisfaction and thwarting of basic psychological needs. The findings suggest the importance of the teacher supporting the satisfaction of students' needs, and avoiding situations that could thwart their needs, in order to promote physical activity through compulsory PE classes.

**Author Contributions:** Conceptualization, R.C.-C. and D.G.-C.; methodology, R.C.-C. and D.G.-C.; validation, J.G.F.-B. and R.C.-C.; formal analysis, R.C.-C. and J.G.F.-B.; investigation, R.C.-C. and A.H.-M.; resources, R.C.-C. and A.H.-M.; data curation, J.G.F.-B. and R.C.-C.; writing—original draft preparation, R.C.-C. and A.H.-M.; writing—review and editing, R.C.-C.; J.G.F.-B.; D.G.-C. and A.H.-M. All authors have read and agreed to the published version of the manuscript.

**Funding:** This research received no external funding.

**Conflicts of Interest:** The authors declare no conflict of interest.

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
