# Peer review of "Need Satisfaction and Need Thwarting in Physical Education and Intention to be Physically Active"

_sustainability, doi:10.3390/su12187312_

Round 1

Reviewer 1 Report

This reviewer feels that minor changes should be made to it.

The abstract and the introduction of the manuscript are adequate, but it would be necessary to include an update of the revised sources, only 16% of the references (10 of 61) are from the last 5 years. The thematic news index, or Price index, must be higher than 30%, so it is advisable to increase the number of citations and references to relevant works of the last 5 years, both in the introduction and in the discussion.

The Method section (participants, instruments, procedure and data analysis) is correct, although it would be necessary to specify more details of the sampling used and the context of the participants, in order to know their characteristics. This detail is important as it is a sample that responds to a non-normal distribution in view of the data provided, despite being 480 cases.

The results presented and the discussion of them is pertinent, just remember that the sources used in the discussion should be updated with works from relevant publications in recent years.

The Final model for the prediction of intention to be physically active in secondary students is adequate and provides useful evidence.

There is some self-citation of the authors, but it does not present conflict as it is related to one of the instruments used.

The reviewer suggested bibliographic update, as well as the inclusion of the characteristics of the sample used (probabilistic or non-probabilistic, or indicate what type it is).

Author Response

First of all, we would like to thank you for your appreciation about our research. We also believe that your considerations can contributed to improve the work.

The reviewer suggested bibliographic update, as well as the inclusion of the characteristics of the sample used.

Done.

In relation to the references, current studies related to the topic of this research have been included. List of the new references added to the manuscript:

6. Trigueros, R.; Aguilar-Parra, J.M.; López-Liria, R.; Rocamora, P. The dark side of the self-determination theory and its influence on the emotional and cognitive processes of students in physical education. International journal of environmental research and public health. 2019, 16 (22): 4444

11. Luis, C.; Pires, A.; Borrego, C.; Duarte-Mendes, P.; Teixeira, D.S.; Moutão, J.M., Monteiro, D. Motivational determinants of physical education grades and the intention to practice sport in the future. Plos one. 2019, 14 (5): e0217218

16. Brooks, J., Huck, G.; Iwanaga, K.; Chan, F.; J Wu, J.; Finnicum, C.; Brinck, E; Estala-Gutierrez, V. Towards an integration of the health promotion models of self-determination theory and theory of planned behavior among people with chronic pain. Rehabilitation Psychology. 2018. 63, 4, 553.

19. Krijgsman, C.; Vansteenkiste, M.; van Tartwijk, J.; Maes, J.; Borghouts, L.; Cardon, G.; Mainhard, T.; Haerens, L. Performance grading and motivational functioning and fear in physical education: A self-determination theory perspective. Learning and Individual Differences. 2017, 1, 202-211.

20. Balaguer, I.; Castillo, I.; Cuevas, R.; Atienza, F. The importance of coaches’ autonomy support in the leisure experience and well-being of young footballers. Frontiers in Psychology. 2018, 9, 840.

24. Sangwook,K.; Lee, K.; Kwon, S. Basic psychological needs, exercise intention and sport commitment as predictors of recreational sport participants exercise adherence. Psychology & Health. 2019, 1-17.

28. Fernández-Espínola, C.; Almagro, B.J.; Tamayo-Fajardo, J.A.; Sáenz-López, P. Complementing the Self-Determination Theory With the Need for Novelty: Motivation and Intention to Be Physically Active in Physical Education Students. Frontiers in Psychology. 2020, 11.

29. Carrón, A.; Lobato, S.; Batista, M.; Leyton, M.; Jiménez-Castuera, R. Prediction of intent to be active physically through the theory of self-determination. Motricidade. 2017, 13, 179.

34. González, L.; Castillo, I.; Balaguer, I. Exploring the Role of Resilience and Basic Psychological Needs as Antecedents of Enjoyment and Boredom in Female Sports. Revista de Psicodidáctica. 2019. 24, 2, 131-137.

61. Li, C.; Wang, C. K. J.; Pyun, D. Y. Impacts of talent development environments on athlete burnout: a self-determination perspective. Journal of Sports Sciences. 2017. 35, 18, 1838-1845.

62. Huhtiniemi, M.; Sääkslahti, A.; Watt, A; Jaakkola, T. Associations among basic psychological needs, motivation and enjoyment within finnish physical education students. Journal of Sports science & Medicine. 2019, 18, 2, 239.

68. Jaakkola, T.; Yli-Piipari, S.; Barkoukis, V.; Liukkonen, J. Relationships among perceived motivational climate, motivational regulations, enjoyment, and PA participation among Finnish physical education students. International Journal of Sport and Exercise Psychology. 2017,15, 3, 273-290.

Also, the reference “10. Hagger, M.S.; Chatzisarantis, N.L.D.; Culverhouse, T.; Biddle, S.J.H. The processes by which perceived autonomy support in PE promotes leisure-time physical activity intentions and behavior: A trans-contextual model. Journal of Educational Psychology. 2003, 95, 784-795. doi: 10.1037/0022-0663.95.4.784” has been deleted.

In this way, we comply with the reviewer's suggestion, including at least 30% of references from the last five years.

On the other hand, regarding the characteristics of the sample used, in the line 103, we had included the sentence: “selected by convenience of two public schools of the región of Castilla-La Mancha (Spain).” We verified that the distribution of the sample data was non-normal. Due the lack of normality in the data, in the data analysis, the maximum-likelihood estimation method with bootstrapping was used. The procedure of bootstrapping affords robust standard errors estimated in these case

Reviewer 2 Report

The reviewed Paper is settled in the educational context and examines the influence of need satisfaction and need thwarting on the intention to be physically active. The authors introduce a theoretical model and test this model in a large sample of 480 students between 12 and 18 years old. Collected data is analyzed in the framework of path analysis. Empirical results highlight the importance of supporting the satisfaction of students’ needs and avoiding situations that could thwart these needs.

I really enjoyed reading this article and thank the authors for the opportunity to review their interesting paper. The topic is important and settled in the educational context. The manuscript is generally well written and structured. For example, chapter 2.3 “Measures” was really good introduced to the reader. As a statistician, I could imagine the variables in my mind and thought about their roles in the authors’ model (e.g., exogenous variable and so on). Later, Figure 1 appeared and visualized my thoughts. Honestly, great work!

However, I do have some concerns with the presentation of the theoretical background as I present in the section below. Besides this, I have some comments/suggestions that I hope will help the authors to further develop this line of work:

  1. Introduction (page 1, line 34): I felt a little bit lost about the construct “physical education (PE)”. What is meant by this? Is it normal sports education in school? Is it a sport club in the afternoon? I searched for additional information about PE in the paper, but I could not find any examples. Thus, I urge the authors to give more information about the construct PE and provide examples to the reader. Perhaps it is useful to introduce PE in a sub-chapter.
  2. Introduction (page 1, line 37): The same is true for “intention to practice physical activity” which is later used as dependent variable in the authors’ model. Please give examples that clarify “physical activity”. Is it running? Is it jumping? Is it soccer?
  3. Introduction (page 2, line 68): Please use the symbol “&” only in brackets and not in the full text. So please write “Bartholomew, Ntoumanis, Ryan, and Thørgersen-Ntoumani“ and not “Bartholomew, Ntoumanis, Ryan, & Thørgersen-Ntoumani” in the full-text.
  4. Chapter 2.3 “Measures” (page 3, line 112): Sometimes “e.g.” without a comma and sometimes “e.g.,” with a comma. Please keep a consistent writing style. Use the latter version only (“e.g.,” with a comma).
  5. Chapter 2.3 “Measures” (page 3, line 139): A lot of commas appear in this and in the following sentence. Please check whether all commas are necessary.
  6. Chapter 2.3 “Measures” (page 4, line 149): The authors cite literature but missed to link this to the references (“De Francisco et al. (2009) found adequate […]”). Please correct this and insert a number that links this citation to the references.
  7. Chapter 2.4 “Data Analysis” (page 4, line 177): Please do not write “[…] if zero stands into the confidence interval, […]” but rather “if zero is included into the confidence interval, […]”. I think the latter version sounds quite better.

Author Response

First of all, we would like to thank you for your appreciation about our research. We also believe that your considerations can contributed to improve the work.

1. Introduction (page 1, line 34): I felt a little bit lost about the construct “physical education (PE)”. What is meant by this? Is it normal sports education in school? Is it a sport club in the afternoon? I searched for additional information about PE in the paper, but I could not find any examples. Thus, I urge the authors to give more information about the construct PE and provide examples to the reader. Perhaps it is useful to introduce PE in a sub-chapter.

Done. Clarification on the term and its meaning has been added in this article. Pag. 1 line 35.

2.- Introduction (page 1, line 37): The same is true for “intention to practice physical activity” which is later used as dependent variable in the authors’ model. Please give examples that clarify “physical activity”. Is it running? Is it jumping? Is it soccer?

Done. Clarification on the term and its meaning has been added in this article. Pag. 1, line 38-39

3.- Introduction (page 2, line 68): Please use the symbol “&” only in brackets and not in the full text. So please write “Bartholomew, Ntoumanis, Ryan, and Thørgersen-Ntoumani“ and not “Bartholomew, Ntoumanis, Ryan, & Thørgersen-Ntoumani” in the full-text.

Done. Line 38.

4.- Chapter 2.3 “Measures” (page 3, line 112): Sometimes “e.g.” without a comma and sometimes “e.g.,” with a comma. Please keep a consistent writing style. Use the latter version only (“e.g.,” with a comma).

Done. In line 114 and following

5.- Chapter 2.3 “Measures” (page 3, line 139): A lot of commas appear in this and in the following sentence. Please check whether all commas are necessary.

Done. Now in pag. 4, lines 141-143.

6.- Chapter 2.3 “Measures” (page 4, line 149): The authors cite literature but missed to link this to the references (“De Francisco et al. (2009) found adequate […]”). Please correct this and insert a number that links this citation to the references.

Done. Now line 151. Referencce number 50.

7.- Chapter 2.4 “Data Analysis” (page 4, line 177): Please do not write “[…] if zero stands into the confidence interval, […]” but rather “if zero is included into the confidence interval, […]”. I think the latter version sounds quite better.

Done. Rewritten, now line 180-181